# Allantoin: A Potential Compound for the Mitigation of Adverse Effects of Abiotic Stresses in Plants

**DOI:** 10.3390/plants12173059

**Published:** 2023-08-25

**Authors:** Rasleen Kaur, Jipsi Chandra, Boby Varghese, S. Keshavkant

**Affiliations:** 1School of Studies in Biotechnology, Pt. Ravishankar Shukla University, Raipur 492 010, India; rasleen.kaur000@gmail.com (R.K.); skeshavkant@gmail.com (S.K.); 2Center for Basic Sciences, Pt. Ravishankar Shukla University, Raipur 492 010, India; jipsi.biotech@gmail.com; 3Centre for Academic Success in Science and Engineering, University of KwaZulu-Natal, Durban 4001, South Africa

**Keywords:** abscisic acid, allantoin, antioxidants, mutants, reactive oxygen species, ureide metabolism

## Abstract

Stress-induced alterations vary with the species of plants, the intensity and duration of the exposure, and stressors availability in nature or soil. Purine catabolism acts as an inherent defensive mechanism against various abiotic stresses and plays a pivotal role in the stress acclimatisation of plants. The intermediate metabolite of purine catabolism, allantoin, compensates for soil nitrogen deficiency due to the low carbon/nitrogen ratio, thereby maintaining nitrogen homeostasis and supporting plant growth and development. Allantoin accounts for 90% of the total nitrogenous compound in legumes, while it contributes only 15% in non-leguminous plants. Moreover, studies on a variety of plant species have reported the differential accumulation of allantoin in response to abiotic stresses, endowing allantoin as a stress modulator. Allantoin functions as signalling molecule to stimulate stress-responsive genes (*P5CS*; pyrroline-5-carboxylase synthase) and ROS (reactive oxygen species) scavenging enzymes (antioxidant). Moreover, it regulates cross-talk between the abscisic acid and jasmonic acid pathway, and maintains ion homeostasis by increasing the accumulation of putrescine and/or spermine, consequently enhancing the tolerance against stress conditions. Further, key enzymes of purine catabolism (xanthine dehydrogenase and allantoinase) have also been explored by constructing various knockdown/knockout mutant lines to decipher their impact on ROS-mediated oxidative injury in plants. Thus, it is established that allantoin serves as a regulatory signalling metabolite in stress protection, and therefore a lower accumulation of allantoin also reduces plant stress tolerance mechanisms. This review gives an account of metabolic regulation and the possible contribution of allantoin as a photo protectant, osmoprotectant, and nitrogen recycler to reduce abiotic-stress-induced impacts on plants.

## 1. Introduction

Environmental stresses are unpredictable, irregular, and ever-changing. Plants are exposed to several complex environmental variables, including temperature, radiation, precipitation, humidity, wind, and soil factors. When a plant experiences less/more than optimum environmental conditions (stress), either through climatic change or human interference, this ultimately affects its survival [1]. Abiotic-stress-induced injuries result in stress-specific responses through distinct modes, and irrespective of the type of stress factor plants elicits a universal response mechanism [2]. Plants can evoke a myriad of responses (morphological, physiochemical, and molecular) to oscillating environmental (stress) conditions [3]. In general, stress conditions stimulate the generation of reactive oxygen species (ROS) such as hydrogen peroxide (H_2_O_2_), singlet oxygen (^1^O_2_), hydroxyl radical (•OH), superoxide (O_2_^•−^) anion, and cytotoxic compounds like methylglyoxal (MG), which disturbs cellular redox homeostasis [4]. The generation of ROS is unavoidable, even under optimal conditions. During normal cellular metabolism, plants can produce 240 μM s^−1^ and 0.5 μM of O_2_^•−^ and H_2_O_2_, respectively. However, abiotic stresses can accelerate ROS generation up to 720 μM s^−1^ O_2_^•−^ and 5–15 μM H_2_O_2_ [5].

Plants employ inherent or acquired extrinsic and intrinsic tolerance or avoidance strategies to resist undesirable environmental conditions. Extrinsic defence strategies provide the first line of defence. Cell walls, thick cuticles and biologically active tissues such as trichomes are a few morphological structures that serves as an external barrier when plants are exposed to stressors [6]. Additionally, plants have robust inherent defence strategies for the tolerance or repair of stress-induced damages. They continuously foster robust machinery to fight against the injurious effects of ROS by using enzymatic antioxidants including superoxide dismutase (SOD), catalase (CAT), ascorbate peroxidase (APX), glutathione reductase, monodehydroascorbate reductase, dehydroascorbate reductase, glutathione peroxidase (GPX), glutathione-S-transferase, and non-enzymatic antioxidants including ascorbic acid, glutathione (GSH), tocopherol, phenolic acids, carotenoids, non-protein amino acids, etc. [7]. Earlier in vitro and in vivo studies have revealed that various osmolytes and osmoprotectants like proline, intracellular complex formation, or the chelation of metal ions by secreting organic acids, phytochelatins, polysaccharides, and metallothioneins are involved in cellular protection through the elimination of excess ROS, osmoregulation, and the stabilisation of protein and membrane structures. These metabolites also play various signalling and regulatory roles in stress-adaptation-mediated plant responses [8]. 

Plants responses to stressors are not only limited to extrinsic and intrinsic strategies, but also include alterations in various indispensable metabolic pathways. Among the potential metabolites that protect plants from oxidative damage against different stressors are ureides, which are intermediate metabolites synthesised through the purine catabolism pathway by the oxidative degradation of purines (Figure 1). Ureides are a collective term used to denote allantoin and its immediate hydrolysed form, allantoate. These are nitrogen-rich compounds that support plant growth, reproduction, and nitrogen recycling. The higher nitrogen-to-carbon ratio in ureides makes them efficient nitrogen transporter molecules through the xylem, majorly in legumes. The ureides have been implicated as antioxidants in ureidic legumes, which participate in ROS scavenging during early seedling development. Higher ureide metabolism under stress conditions has revealed their protective roles [9,10,11,12,13] (Figure 1). For example, the metabolic profiling of *Arabidopsis thaliana* under heat and cold stress exhibited higher concentrations of allantoin [14]. The present review mainly focused on providing deeper insights into the differential accumulation and interaction of allantoin-mediated responses in plants towards ameliorating heavy metals (HM), nutrient deficiency, drought, salinity, irradiance, darkness, and ultraviolet (UV)-radiation-induced metabolic impairments.

## 2. Synthesis and Transportation of Allantoin

Allantoin (C_4_H_6_N_4_O_3_), also called glyoxyldiureide (diureide of glyoxylic acid) or 5-ureidohydantoin, is a heterocyclic nitrogen-rich compound universally present in plants and commercially exploited in both cosmetic and pharmaceutical industries [15]. It is an intermediate metabolite of purine catabolism and follows a common synthesis pathway in all plants. The metabolism of ureide (allantoin and allantoate) in the *Arabidopsis thaliana* involves seven enzymatic reactions which take place in three different cell organelles: the peroxisome, cytosol, and endoplasmic reticulum (ER) [16]. Purine nucleotides (guanine and adenine) are precursor molecules to instigate the biosynthesis of allantoin either via the de novo pathway in leguminous plants or the salvage pathway in non-leguminous plants. Allantoin biosynthesis starts with adenosine monophosphate (AMP) conversion to inosine monophosphate (IMP) via the activity of the enzyme AMP deaminase (AMPD). The conversion of IMP to xanthine monophosphate (XMP) in the de novo synthesis of guanine nucleotide is the rate-limiting step catalysed by the enzyme IMP dehydrogenase. Inosine monophosphate can further lead by multiple pathways for producing inosine, XMP, or guanine monophosphate (GMP) (via XMP conversion). Inosine converts to hypoxanthine, which is acted on, in turn, by xanthine dehydrogenase to synthesise xanthine. Xanthine is also formed from the other purine GMP by deamination [17,18]. Within the cytoplasm, the xanthine dehydrogenase (XDH) enzyme catalyses the conversion of xanthine into urate. Anion-selective channels are present on the peroxisomal membrane to import urate to peroxisome from the cytoplasm for its further oxidation to 5-hydroxyisourate (HIU) by urate oxidase (UO). This 5-hydroxyisourate is converted via 2-oxy-4-hydroxy-4-carboxy-5-ureido-imidazoline (OHCU) to allantoin, catalysed by bifunctional enzyme allantoin synthase. Allantoin is then imported to the ER, where the allantoinase (ALN) converts it into allantoate. Allantoate degradation may follow two pathways; either allantoate amidohydrolase, or allantoate amidinohydrolase. Allantoin amidohydrolyase (AAH) catalyses the conversion of allantoin to allantoate via the hydrolysis of its internal amide bond. In the allantoate amidohydrolase pathway, allantoate is hydrolysed to form urea and ureidoglycolate. Ureidoglycolate is further broken down to glyoxylate and urea. Urea formed in this pathway is further hydrolysed to ammonia (NH_3_) and carbon dioxide (CO_2_) by the action of the urease enzyme. Allantoate is hydrolysed to CO_2_, NH_3_, and ureidoglycine in an alternative pathway. Ureidoglycine is an unstable compound, and therefore quickly deaminated, either spontaneously or enzymatically, to produce ureidoglycolate. Allantoin amidohydrolyase finally acts upon ureidoglycolate to produce NH_3_, CO_2_, and glyoxylate. Allantoin amidohydrolase is suggested to be involved in the last step of ureide catabolism [16,19,20] (Figure 1). Ammonia released by ureide metabolism supports plant growth and development. 

Allantoin exists in two stereospecific isomers, namely R-allantoin and S-allantoin. The most prevalent physiologically active type of allantoin in plants is S-allantoin [21]. The transportation of allantoin to other parts of the plant follows either apoplastic or symplastic pathways. Pelissier et al. [22] studied allantoin transport in *Phaseolus vulgaris* nodules involving PvUPS1, an allantoin transporter. In the symplastic pathway, allantoin is transported through the inner cortex and endodermis through plasmodesmatal connections [23]. On the other side, apoplastic allantoin transport occurs from nodules to the xylem. Casparian strips hinder allantoin transport, which is overcome by the involvement of secondary active transporters like ureide permease (UPS), bound to the plasma membrane [24] (Figure 1). This UPS has characteristic ten-membrane-spanning transmembrane domains with the “Walker A” motif situated inside the cytoplasm. 

Moreover, many UPS transport coding genes have been identified in different plant species, such as *Arabidopsis thaliana* [25], *Phaseolus vulgaris* [26], *Glycine max* [27], and *Oryza sativa* [28]. In *Arabidopsis thaliana*, various types of UPS (AtUPS1, AtUPS2, AtUPS3, AtUPS4, AtUPS5l, and AtUPS5s) were identified and compared for their relative affinity for allantoin, establishing which among the AtUPS2 located at root stele possesses highest affinity. A higher expression of UPS within *Arabidopsis thaliana* has been found to be positively correlated with the increased transport of nitrogen from senescent leaves to younger leaves for their growth [28,29]. The over-expression of the OsUPS1 transporter in *Oryza sativa* leads to a higher accrual of allantoin in shoot tissues. Similarly, OsUPS1 knockdown lines of *Oryza sativa* revealed a dramatic reduction in the accumulation of allantoin in shoots [28]. Primarily, UPS transporters aid in transporting the allantoin from its site of synthesis to the organs to be utilised as a nitrogen source in a range of plant species [13]. However, studies have revealed the role of AtUPS1 in the transport of exogenously applied allantoin in plants [25].

## 3. Defensive Strategies of Allantoin against Abiotic Stressors

### 3.1. Salinity Stress

Plants abilities to withstand salinity stress and uphold their growth potential rely on nitrogen metabolism. Allantoin has been studied as one of the salt-responsive metabolic indicators in *Oryza sativa* roots. The metabolic profiling of salt-tolerant (G58, G1710, and IR64) and salt-sensitive (G45 and G52) *Oryza sativa* lines demonstrated that the allantoin is an active metabolite under salinity stress [30]. Intriguingly, both salt-sensitive and salt-tolerant species exhibit comparable metabolic pathways when plants perceive salt signals, with key variations in the number of metabolites present.

Salinity stress has negative impacts on dry weight, nitrogen content, and the rate of nitrogen fixation in legumes due to altered nodule structures and reduced nitrogen metabolism capability. Salt treatment {25, 50 and 100 mM of sodium chloride (NaCl)} applied to *Phaseolus vulgaris* exhibited a reduced dry weight (25%) and decreased nitrogenase activity (50%) following glutamine synthetase (30%) and glutamate synthase (50%) [31]. Additionally, activities of XDH and UO, key enzymes of purine catabolism, declined significantly along with salt concentration [11]. Moreover, knockout, knockdown, and stress-inducible ALN mutants-based phenotypic studies emphasised the importance of allantoin accumulation to overcome salinity stress. The application of 150, 200, and 250 mM NaCl solution imposed visual symptoms such as chlorosis and growth reduction in *Arabidopsis thaliana* [24]. Salt treatment exhibited enhanced expression/mRNA levels of UO, while reducing *AtALN*. The *AtUPS5* expression associated with allantoin long-distance transport has been seen to be up-regulated by salt treatment [32]. 

Moreover, mutants with low *AtUPS5* expression levels revealed sensitivity to salt stress. Additionally, these mutants were inefficient in using externally applied allantoin as a nitrogen source. Irani & Todd [11] demonstrated that the ALN mutant of *Arabidopsis thaliana* accumulated higher amount of allantoin, while a significantly lower level of ROS (H_2_O_2_ and O_2_^•−^) was observed as compared to that estimated in wild-type *Arabidopsis thaliana* (Figure 2). In another study, *Arabidopsis thaliana* was exposed to 50 mM, 75 mM, and 100 mM NaCl, which exhibited phenotypic changes like reduced root growth, smaller leaves, and five-fold increase in allantoin content with lower expression of *ALN*, *AAH*, and *ALNS* genes in saline-treated seedlings. On the other hand, mutant seedlings revealed an enhanced root length and the appearance of well-developed leaves compared to wild-type plants when cultivated under saline stress. Improved salt tolerance in mutants suggests that allantoin accumulation affects salt stress tolerance throughout the vegetative growth period, which manifests increased biomass accumulation and photosynthetic efficiency [33]. 

Additionally, the exogenous application of allantoin (0.1mM and 1mM) provided salt tolerance to *Arabidopsis thaliana* seedlings, which might be associated with higher levels of allantoin accumulation than usual [33]. You et al. [34] developed transgenic *Arabidopsis thaliana*, with enhanced salt tolerance. These plants acquired salt tolerance after the incorporation of the *XDH* gene of *Vitis vinifera*, named *VvXDH*. This gene was seen to be over-expressed in transgenic plants, consequently increasing allantoin levels. In response to NaCl concentrations of 150, 200, and 250 mM, the *VvXDH* transgenic lines VT-1, VT-2, and VT-3 showed germination values of 50%, 47.22%, and 33.33%, respectively, while wild plants showed 5.56% germination. Enhanced allantoin levels lowered the water loss and malondialdehyde (MDA) content, along with enhancements in chlorophyll (Chl) and proline accumulation in transgenic plants compared with those determined in the wild type (Figure 2). 

In a similar study, when allantoin (0.01 mM) was added to a medium containing NaCl (100 mM), it promoted salinity tolerance in the salt-sensitive genotype IR-29 of *Oryza sativa*. Allantoin supplementation has been found to enhance shoot and root length along with biomass in IR-29 seedlings [35]. Stress-responsive gene expressions are considered to be a hallmark of plant adaptation to stress. *AtP5CS* gene-encoding delta-1-pyrroline-5-carboxylate synthase, an enzyme of proline biosynthesis, was seen to be highly regulated in the *VvXDH* over-expressing *Arabidopsis thaliana* under salt stress. The accumulation of proline helped in maintaining the osmotic equilibrium between the extracellular and intracellular environment, thereby protecting membrane integrity, which promoted salt tolerance. Allantoin has also been shown to be involved in regulating ion balance under salt stress by compensating the absorption of sodium, potassium, calcium, and magnesium ions to maintain ion homeostasis. Additionally, the activities/expressions of ROS scavenging enzymes/genes *AtSOD*, *AtPOD*, and *AtCAT* were significantly up-regulated in transgenic plants under salt stress. In conclusion, elevated XDH activity and enhanced allantoin and proline content stimulated the ROS scavenging system in the transgenic *Arabidopsis thaliana* [34]. 

Polyamines (PAs) are low-molecular-weight aliphatic amines that exist in free forms, including putrescine (Put) and/or spermine (Spm) and spermidine [36]. They participate in regulating various physiological processes and are actively involved in responses to biotic and abiotic stresses. Allantoin may improve ion homeostasis and the antioxidant system in salinity conditions by enhancing PAs’ accumulation. In plants, arginine decarboxylase (ADC) and S-adenosylmethionine decarboxylase (SAMDC) are involved in the biosynthesis of PAs, while diamine oxidase (DAO) and polyamine oxidase (PAO) are responsible for the catalytic degradation of the PAs [37,38]. Exogenous allantoin pretreatment modulated the activities of these enzymes in *Beta vulgaris* seedlings subjected to salt stress (300 mM). The activity of ADC in the leaves of *Beta vulgaris* increased with increasing allantoin concentrations (0.01, 0.1, and 1 mM), leading to a higher accumulation of Put. The external application of allantoin (0.1 mM) improved the leaf, root, and plant biomass by 22%, 60.56%, and 63.44%, respectively, compared to the control [39]. Allantoin improved the salt tolerance of *Beta vulgaris* with a higher accumulation of Spm and Put. This implies that the accumulation of PAs is mediated by allantoin that may help to improve salt tolerance in plants. 

Studies have revealed that the MicroRNAs (miRNAs)-guided gene regulation may influence how plants react to salinity stress. MicroRNAs are a class of short non-coding RNAs (21–24nt) that control the expression of target genes either through post-transcriptional translational inhibition or cleavage [40]. Nishad et al. [35] analysed the effects of exogenously applied allantoin on *Oryza sativa* miRNAs under salinity stress. It is observed that miRNAs, namely os-miR393a, osa-miR414, osa-miR530, and osamiR818a, were down-regulated under salinity stress [41]. Allantoin supplementation reversed the expression patterns of these miRNAs in the salt-sensitive genotype IR-29 of *Oryza sativa*. Therefore, allantoin might regulate salinity stress tolerance through the interaction with miRNAs. 

### 3.2. Drought Stress

The proton nuclear magnetic resonance-based metabolic profiling of drought-sensitive (DM50048) and drought tolerant (NA5009RG) *Glycine max* genotypes revealed contrary trends of ureide (allantoin) accumulation. The species type, genotype, duration of exposure, and developmental stage are a few contributing factors that determine the tolerance capacity or sensitivity of species towards water stress. Ureide accumulation has occurred in leaves, shoots, and roots of drought-sensitive *Glycine max* cultivars [42]. To gain deeper insight into molecular, physiological, and metabolic changes, *Triticum aestivum* was exposed to drought conditions. Under progressive drought conditions, allantoin levels were enhanced by 120-fold, indicating the involvement of ureide metabolism as a survival mechanism under severe drought stress [43]. Similarly, the metabolic profiling of *Cicer arietinum* revealed allantoin accrual following long-term drought stress [44]. Likewise, a comparative study of two *Triticum aestivum* genotypes revealed a significant (up to 29-fold) increase in allantoin concentration under drought conditions [45]. 

Genotypic variation in response to drought stress is associated with the biochemical and transcriptional regulation of ureide metabolism [46]. Plants undergo a complex metabolic regulation/adjustment in response to water stress conditions. Under a scarcity of water, plants limit nitrogen fixation either as an outcome of low photosynthate supply or a decrease in oxygen flux or an accumulation of nitrogenous compounds within the nodules. Inhibition in nitrogen fixation could also be attributed to ureide accumulation. The inactivity of AAH (an allantoate degrading enzyme) contributes to ureide accumulation in various plant tissues [9]. Ureidic legumes systemically induce the accumulation of ureides (allantoate) in roots, shoots, and leaves, but are limited to nodules in drought-stressed *Phaselous vulgaris* plants [47]. Physiological studies on drought-sensitive genotypes showed a decline in the relative water content and leaf dry weight. Ureide accumulation in response to water limitations was regulated at the molecular level by the inhibition of allantoate synthesis. The expression of the *UO* transcript was highly enhanced in plant tissues. This was in parallel to *ALN* mRNA expression in roots, shoots, and leaves of drought-stressed tissues of *Phaselous vulgaris* [47]. 

Watanabe et al. [48] developed mutant (xdh4 and xdh5 in the T4 generation) cell lines of *Arabidopsis thaliana* containing suppressed XDH to unveil its sensitivity towards drought tolerance. Xanthine dehydrogenase suppression via RNA interference resulted in pleiotropic phenotypes with deferred development, impaired fertility, and early senescence. The accumulation of ROS is accelerated in plants under water stress. Drought-hypersensitive phenotypes exhibited reduced plant growth with significantly enhanced cell death and ROS (H_2_O_2_) accumulation. The exogenous supplementation of uric acid (25 µM) restored the impaired growth in phenotypes and reduced susceptibility to drought. Uric acid has long been recognised for potentially scavenging ROS in both in vitro and in vivo conditions. In another similar work, Watanabe et al. [9] studied how knocking down XDH severely affected the survival of *Arabidopsis thaliana* via a decreased tolerance to O_2_^•−^ mediated oxidative stress. Under progressive drought conditions, the concomitant transcriptional up-regulation of proline-biosynthetic enzymes with the down-regulation of proline-degrading enzymes mediates higher proline accumulation. Plants induce the accumulation of cellular protectant proline as a stress response. Knockdown plants exhibited lower mRNA transcript levels of the *P5CS1* gene, encoding the proline biosynthesis enzyme. Externally applied allantoin (10 to 100 nM) and urate elicited *P5CS1* transcript expression 3.5- and 4.2-fold, respectively. Both the ALN-knockout mutant and exogenous allantoin resulted in the activation of the abscisic acid (ABA) pathway, suggesting that a regulatory role of allantoin in stress protection is mediated by the activation of the ABA metabolism [10,43]. 

### 3.3. Heavy Metals

#### 3.3.1. Cadmium 

Allantoin accrual acts as an indicator of oxidative stress conditions, and the protective function of allantoin may be caused by its direct or indirect influence on ROS accumulation [49]. Studies have demonstrated that cadmium (Cd) stress initiates ureide metabolism and leads to the accumulation of allantoin and allantoate to alleviate the detrimental effects of stress and confer tolerance in plants [9,11] (Figure 2). A study was conducted on wild-type ALN-negative (aln-3) mutants of *Arabidopsis thaliana* to demonstrate the impacts of Cd exposure on ureide metabolism and its role in stress resistance. Wild-type plants showed a discolouration and reduction in the leaf area and decreased seedling growth, with an increase in the concentration of Cd. Moreover, inhibition in germination was also observed after the treatment of 200 mM Cd. Contradictorily, the aln-3 mutants grown on similar concentrations of Cd showed more resistance, with higher root length, increased biomass accumulation, and seed germination (Figure 2). Furthermore, Nourimand & Todd [50] observed nearly two-fold increases in allantoin content in response to 1000 μM Cd in aln-3 mutants (Table 1). An inverse relation was observed between the concentration of Cd and allantoin, which was attributed to an elevated basal level of allantoin to reduce adverse impacts of Cd exposure, subsequently conferring Cd tolerance in plants. 

Urate oxidase, ALN and AAH are essential enzymes of ureide metabolism. Urate oxidase is responsible for allantoin accumulation, and ALN degrades allantoin into NH_3_. Alterations in the transcription levels of allantoin synthesising (*ALNS*, *XDH*, *UO*) and degrading (AAH, ALN) enzymes have been determined in response to Cd stress [11]. Gene expression studies showed that the transcripts of ALN and AAH decreased in Cd-treated plants compared with the non-treated controls. A significant increase in allantoin concentration was concomitant with lower uric acid associated with the up-regulation of *UO* with increased Cd concentration, suggesting that the Cd influences allantoin accumulation [50]. Increased allantoin supported better growth and higher Cd uptake (Figure 2). The aln-3 mutants stored Cd in the leaves, which did not negatively impact their photosynthetic efficiency, while downregulation in *UO* and *ALN* was responsible for decreased allantoin levels in the roots of wild-type plants [51] (Table 1). 

Roots are the primary site to be exposed to various soil pollutants, including HMs. Nourimand & Todd [51] studied the role of allantoin in the fortification of roots against Cd stress. Overproduction of ROS was observed in the roots of both mutant and wild-type *Arabidopsis thaliana*. Mutants accumulated higher allantoin and lower levels of ROS than those determined in the roots of wild-type plants under the same set of conditions. The activities of enzymatic (SOD, glutathione reductase, CAT, and APX), and non-enzymatic (dehydroascorbate and reduced/oxidised glutathione) antioxidants were also recorded up to an extent under Cd stress in *Brassica juncea* [52]. Allantoin in aln-3 mutants alleviated Cd toxicity by eliciting antioxidants, which maintained the ROS levels and improved growth attributes [51]. Therefore, allantoin accumulation can be considered a sign of oxidative stress in plants as ROS accumulation influences allantoin activation, and through this plants achieve their defensive properties. 

Additionally, a ureide transporter UPS in root cells showed a high affinity for allantoin and regulated its transportation to other plant parts [24]. However, a higher amount of Cd was absorbed from the soil by aln-3 mutants, which restricted the translocation of allantoin. This approach protects shoots from Cd-induced oxidative damage and rescues the imperative function of photosynthetic apparatuses, even under stress [53]. The defensive role of allantoin in early phases of seed germination and seedling growth indicated that this ureide may have stage-specific functions with different phases of plant development.

Furthermore, the impact of the exogenous application of allantoin (10, 100, 1000 μM) on *Cucumis sativus* experiencing Cd (5–15μM) stress was studied by Dresler et al. [54] (Table 1). Allantoin considerably reversed Cd-stress-induced toxicities in treated plants in a concentration dependent manner. It enhanced the biomass of roots and shoots, the leaf area, and photosynthetic pigments in stressed plants. Allantoin supplementation suppressed the H_2_O_2_ levels, which was associated with improved enzymatic (CAT and APX) and non-enzymatic (GSH and ascorbic acid) antioxidants (Figure 2). This demonstrates that allantoin may be related to non-enzymatic antioxidants. In another study, the effect of exogenously applied allantoin was observed on wild ecotype and allantoinase-overexpressed (ALNox) lines of *Arabidopsis thaliana* under Cd stress. Mutant seeds showed more sensitivity to Cd stress, exhibiting a positive correlation between the concentration of allantoin and tolerance to Cd, whereas the wild-type exhibited improved germination, root elongation, and antioxidant activities [55] (Table 1). Another study focused on the influence of additional sources of nitrogen (nitrate, urea, or allantoin) on Cd toxicity and microbial activity and concluded that supplemented allantoin was metabolized by plants/microbes into urea, which significantly elevated rhizospheric microbial activity, thereby mitigating Cd toxicity in *Cucumis sativus* [56].

**Table 1 plants-12-03059-t001:** Role of allantoin in heavy metal detoxification and mitigation of abiotic stresses.

S. No.	Stress Factors	Plant Species	Mutants Developed	Inference	Parameters Studied	References
Heavy metal						
1.	Cadmium	*Arabidopsis* *thaliana*	ALN-negative (aln-3)	Shoot allantoin↑, UO activity↑, ALN↓, Proline content↑, ROS levels↓, Antioxidant levels↑	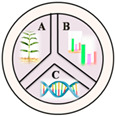	Nourimand and Todd [50]
2.	Cadmium	*Arabidopsis* *thaliana*	ALN-negative (aln-3)	Root Cd level↑, Root biomass↑, UO transcript level↑, ROS levels↓, Antioxidant levels↑	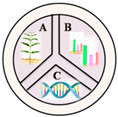	Nourimand and Todd [51]
3.	Zinc/Lead	*Echium* *vulgare*	-	Allantoin levels↑, Metal accumulation in roots↑	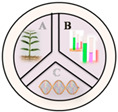	Dresler et al. [57]
4.	Zinc/Lead	*Echium* *vulgare*	-	Allantoin levels↓, Rosmarinic acid↓	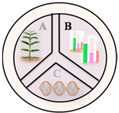	Dresler et al. [58]
5.	Zinc, Lead, and Cadmium	*Echium* *vulgare*	-	Plant biomass↑, Allantoin levels↑, Chlorogenicandrosmarinic acids↑, Total phenolicsand flavonoids↑, Malate and citrate content↑	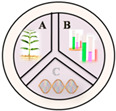	Dresler et al. [59]
6.	Strontium	*Glycine* *max*	-	↑Allantoin levels in roots	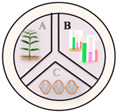	Dresler et al. [60]
7.	Cadmium	*Arabidopsis* *thaliana*	ALNox, abi mutants	Seedling growth↑, Root elongation↑, Antioxidant levels↑	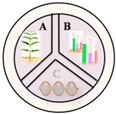	Nourimand and Todd [55]
8.	Cadmium	*Cucumis* *sativus*	-	Shoot biomass↑, Leaf area↑, Citric acid↑, Phytochelatins↓, ROS levels↓, Glutathione and ascorbic acid↑, Photosynthetic pigments↑	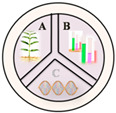	Dresler et al. [54]
9.	Strontium	*Glycine max*	-	↑Allantoin levels in roots	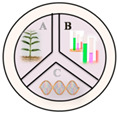	Hanaka et al. [61]
Nutrient deficiency						
10.	Sulphur deficiency	*Arabidopsis* *thaliana*	-	Ureides content↑	* 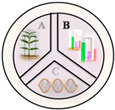 *	Nikiforova et al. [62]
11.	Nitrogen deficiency	*Arabidopsis* *thaliana*	Atxdh1, Ataln, Ataah	Ureides content↑	* 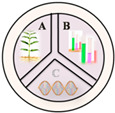 *	Soltabayeva et al. [29]
UV-C Stress						
12.	*-*	*Solanum* *lycopersicum*	-	Allantoin and allantoate content↑, PAL↑, Antioxidants↑, Chlorophyll↑, Soluble protein and carbohydrate↑	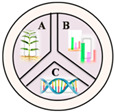	Dawood et al. [63]
13.	UV-C+ Wounding	*Arabidopsis* *thaliana*	Atxdh1	Allantoin↓, ROS levels↑, MDA↑, Shoot fresh weight↓, EL↑, Chlorophyll↓, Senescence gene expression↑, Autophagy↑	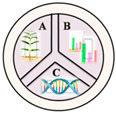	Soltabayeva et al. [64]

abi: abscisic acid (ABA)-insensitive mutants; ALN: allantoinase; ALNox: allantoinase-over-expressed mutants; Ataah: allantoate amidohydrolase; Ataln: allantoinase mutant; Atxdh1: xanthine dehydrogenase1mutant; EL: electrolyte leakage; MDA: malondialdehyde; PAL: phenylalanine ammonia-lyase; ROS: reactive oxygen species; Uo: uricase. A, B, and C in the parameters studied show morphological, biochemical, and molecular studies, respectively. Similarly, arrows up (↑) and down (↓) in the inference column represent enhancement or decline in the activity/content.

#### 3.3.2. Other Heavy Metals 

Exposure to high concentrations of zinc (Zn) and lead (Pb) initiates the development of unique features in plants via the evolution of ecotypes for their survival in extreme environments [65]. In addition to perceptible morphological traits, metallicolous populations (plants that thrive in metal-rich soil) have evolved various biochemical appliances of HM detoxification. Izmailow & Biskup [66] observed considerable distinctions in the contaminant adaptability of *Echium vulgare* populated in metal-polluted and non-polluted regions. Moreover, Dresler et al. [67] documented the morphological and physiological effects of Cd on metallicolous and non-metallicolous populations of *Echium vulgare* and documented genetic diversity in these populations colonising on Pb-Zn waste heaps. The concentration of allantoin in the metallicolous population was 4- and 10-fold higher in shoots and roots, respectively, than those detected in similar tissues of non-metallicolous population [57] (Table 1).

Increases in the concentrations of allantoin and shikonin due to the stressful growth environment indicated that these compounds might play an important role in the enhancement of plant resistivity towards stress factors, including HM of metalliferous areas. Dresler et al. [57,59] determined the responses of metallicolous and non-metallicolous populations of *Echium vulgare* after exposure to chronic multi-metal (Zn, Cd, Pb) and acute Pb (30, 60 mM) and Zn (200, 400 mM) stress (Table 1). The Pb/Zn stress caused enhanced accretions of allantoin, rosmarinic acid, chlorogenic acid, flavonoids, and total phenolics. Moreover, the sequestration of Pb in the root cells, with the translocation of Zn to the shoots, played a crucial role in the increased metal resistance of metallicolous populations under acute Pb/Zn stress. However, phytochemical studies of seeds of *Echium vulgare* from Zn/Pb rich areas showed the low concentration of allantoin [58] (Table 1). Exposure to HM may cause the interruption or even termination of seed germination. The translocation of HM in plant organs declines in the following sequence: roots>leaves>stems>inflorescences>seeds [68]. Metallicolous plants accumulate one to ten times more metals due to their habitation on HM-enriched sites.

The presence of strontium (Sr) beyond the threshold limit (30 ppm) led to a diminution in growth parameters, which was positively correlated with a reduction in cell division [69]. Strontium is also studied as an elicitor for the accumulation of secondary metabolites, including phytoestrogens and allantoin. Dresler et al. [60] showed that the long- term exposure of Sr (0.5–4.0 mM) affected the accumulations of phytoestrogens and allantoin in roots of *Glycine max* (Table 1). A strontium-mediated increase in the concentration of allantoin in roots suggested its significance as a stress mitigation compound [61] (Figure 1, Table 1). Kovacik et al. [70] studied the impact of organic nitrogen supplementation on Sr (100 μM)-induced metabolic changes in *Hypericum perforatum* plants. The addition of nitrogen forms modulated ROS and nitric oxide formation in the root tissues, elucidating the protective action of nitrogen-rich compound allantoin under a stress environment [71]. 

### 3.4. High Irradiance

Studies have indicated increase in the metabolites of ureide metabolism in plants in response to high irradiance. *Eutrema salsugineum*, an extremophile, was identified to possess the capacity to withstand numerous abiotic stressors. Irani et al. [72] chose this species as a model to unveil the allantoin response in plants facing high-intensity irradiance as stress contributor. They observed differences in growth responses and metabolic components under high to medium irradiation intensity. Two different growth irradiances, moderate light (ML) and high light (HL), which corresponded to a photosynthetic photon flux density of 250 and 750 μmol (photon) m^−2^ s^−1^, respectively, were used. Plants were observed to acclimatise with HL and ML with an increase in specific leaf mass. This may happen because the ureide content increased by 2.5-fold in leaf tissues developed under varying irradiance conditions compared to non-irradiated ones [12]. Further, molecular studies demonstrated that the genes involved in both the synthesis and degradation of ureide were induced upon stress exertion. The expression levels of allantoate degradation genes *AAH* and *UGlyAH* were up-regulated in response to HL irradiation.

In another study, the accumulation of ureides in *Arabidopsis thaliana* under high irradiance (HI) and in response to increased photooxidation has been determined by Irani et al. [72]. They compared the responses of ALN-negative mutants and their wild types under HI. Mutants were devoid of allantoin-degrading enzyme (ALN), causing higher levels of allantoin accumulation within the plants, termed aln-3 *Arabidopsis thaliana* mutants. In these mutants, the allantoin content reached a maximum of 0.72 μmol g^−1^ dry mass on the seventh day and diminished thereafter. Moreover, a lesser accumulation of anthocyanin in aln-3 mutants could be related to the lower sensitivity of these to HI. 

Conversely, key genes regulating the synthesis and degradation of allantoin, namely, *UO*, *ALN*, and *AAH*, showed a higher expression in wild-type plants. Thus, the plants showed several morphological changes, such as hastened reproductive development, thickened leaves, reduced Chl content, and increased anthocyanin, imparting a purple colour to the leaves after a certain amount of HI exposure. On the other side, Chl-a and Chl-b content and the quantum yield ratio were also significantly higher in HI-treated aln-3 mutants of *Arabidopsis thaliana* than in the wild types. Constitutively, a higher allantoin content in aln-3 mutants led to a higher tolerance to HI stress (Figure 2).

### 3.5. Dark Stress

Accumulating higher levels of allantoin is one of plants primary defence responses. Castro et al. [73] studied the effect of the photoperiod on the patterns of allantoin accumulation in roots and rhizomes of *Symphytum officinale*. In these plants, stress conditions promoted the remobilisation of resources as an adaptive strategy for environmental adaptation. Contrary to HI, dark conditions with no light also induce plant stress. Dark-induced stress causes the yellowing of leaves due to the breakdown of Chl and chloroplasts [17]. In such conditions, purine catabolism serves as the energy conserving pathway by generating higher carbon/nitrogen ratio molecules. In plants, the molybdenum co-factor and XDH are involved in purine remobilisation. The XDH mediates the conversion of purine catabolism products hypoxanthine and xanthine into uric acid. 

Further, allantoin and allantoate are formed as end products of uric acid degradation (Figure 1). Plants under extreme conditions of extended darkness induce transcripts of purine catabolism. Mutants of *Arabidopsis thaliana* generated by attenuated XDH known as Atxdh1 accumulated higher xanthine, and then underwent premature senescence, in which symptoms of senescence were determined to be pronounced by the enhanced degradation of Chl, cell death, and the up-regulation of ageing-related transcripts. 50% decline in the level of residual Chl was observed in mutant plants. Moreover, the early senescence transcript, *WRKY53* (senescence-related regulatory gene) and *ERD1/SAG15* (senescence-associated gene) [74], and transcripts of genes involved in senescence-associated Chl breakdown [*ACD1* (accelerated cell death1)] and Chl catabolism [*SGN1* (stay-green protein1)] [75] were seen to be highly induced in Atxdh1 mutants under dark stress. Brychkova et al. [17] observed two different forms of XDH, XDH1 and XDH2 in *Arabidopsis thaliana*, and compared them with dark exposure. They observed an about 20-fold increase in the transcripts of XDH1 in *Arabidopsis thaliana* after six days of dark exposure. However, the expression levels of XDH2 transcripts were unaffected under dark stress. Moreover, a rapid decline in the level of XDH1 was seen after the exposure of similar plants to light conditions and it was observed that the XDH1 was normally induced in a normal diurnal physiological response. However, its level rose higher during the extensive dark period. 

Dark conditions provoke the purine catabolic flux. Higher levels of allantoin and allantoate in wild-type plants under dark stress have suggested their role in ameliorating dark-induced responses. Additionally, up-regulation in ureide-synthesising key genes (*AMPD*, *XDH1*, *UO*, and transthyretin-like protein *TLP*) was concomitant with a decrease in the transcripts of ureide utilising enzymes (ALN and AAH) in dark stressed plants. Moreover, the exogenous addition of ureides has also been found to prolong cell viability in *Arabidopsis thaliana* seedlings during extended darkness [17].

### 3.6. Nutrient Deficiency 

Mineral elements absorbed in their inorganic forms are crucial for the growth and development of plants. An abundance of a particular nutrient causes the non-availability of other nutrients or a limitation in the genetic potential of crops, resulting in no increase or even a decrease in yield [76]. In a study, uricase activity and allantoin distribution against ammonium sulphate were investigated by analysing the accumulation of nitrogenous compounds and the expression of the *MsU2* gene in *Medicago sativa* plants. A study suggested that the application of excessive ammonium in legume plants not only affects the distribution of ureides and the ability for nitrogen fixation, but also leads to a decrease in the utilisation of fixed atmospheric nitrogen. The profiling of nitrogen-stressed plants showed higher accumulations of purine metabolites like allantoin, which indicates a degradation of the purine ring to compensate for nitrogen deficiency. Allantoin accounts for 90% of the total nitrogenous compound in legumes, while it contributes only 15% in non-leguminous plants [13]. Nearly 8-fold accumulation of allantoin was determined in nitrogen-stressed *Oryza sativa* compared to the non-stressed one. Such a higher accumulation of allantoin suggested the involvement of this catabolic pathway in nitrogen remobilisation under nitrogen-limiting conditions [77].

Plants respond and adapt to environmental changes by using organic compounds that widely differ in their carbon/nitrogen ratio. The ureides (allantoin and allantoate) have a lower carbon/nitrogen ratio (1/1) and serve as principal compounds for reduced nitrogen transport (using UPS) from root to the shoot in tropical legumes [78]. The two functional allantoinase genes responsible for the degradation of allantoin, *AtALN* and *RpALN*, were over-expressed in the absence of other nitrogen sources, providing evidence for allantoin as an alternative nitrogen source [21]. Under nitrogen deficiency, increased allantoin catabolism contributes towards maintaining nitrogen homeostasis [79]. On the other hand, one study has shown the limited role of allantoin as a nitrogen source in cultured *Coffea arabica* cells [80].

Soltabayeva et al. [64] examined the roles of allantoin and allantoate as nitrogen sources in wild-type *Arabidopsis thaliana* and its xdh1 mutant (Table 1). The xdh1 mutant exhibited reduced soluble protein, organic nitrogen, and early senescence compared to wild-type plants. Moreover, nitrogen deficiency evoked the purine degradation pathway in the senescing leaves to provide nitrogen to young leaves. Nitrogen-stressed mutants showed premature senescence, while nitrogen-supplied mutants not only exhibited the disappearance of premature senescence symptoms but also enhanced the organic nitrogen and soluble protein content. Ureide permease transports ureide from the nodules of legume plants to their shoot regions. Moreover, *AtUPS1* has been seen to be up-regulated in *Arabidopsis thaliana* shoots in response to nitrogen limitations [81]. The induction of the *UPS* transcripts is responsive to low nitrate supply [25]. The expression of the ureide transporter gene was found to be up-regulated in older leaves of low-nitrogen-supplied wild-type *Arabidopsis thaliana* as compared to plants provided with a sufficient amount of nitrogen.

Lee et al. [82] studied the role of allantoin as a source of nitrogen in *Oryza sativa*. The expression levels of two genes involved in ureide metabolism, ALLANTOINASE (*OsALN*) and UREIDE PERMEASE 1 (*OsUPS1*), were seen to inversely regulate the nitrogen status. ALLANTOINASE was rapidly up-regulated under low nitrogen conditions, whereas *OsUPS1* was up-regulated under high nitrogen conditions. Based on the responses of these two genes (*OsALN* and *OsUPS1*), in vivo molecular nitrogen sensors, proALN::ALN-LUC2 and proUPS1::UPS1-LUC2, were developed in transgenic *Oryza sativa* plants. Transgenic *Oryza sativa* plants expressing proUPS1::UPS1-LUC2 showed strong luminescence under high exogenous nitrogen conditions (>1 mM), whereas transgenic plants expressing proALN::ALN-LUC2 exhibited strong luminescence under low exogenous nitrogen conditions (<0.1 mM). Proteomic studies have shown that nitrogen status has a differential effect on the growth and adaptation of *Oryza sativa*. Compared with normal nitrogen conditions, 291 and 211 differentially abundant proteins were identified under low and high nitrogen conditions, respectively [83].

Sulphur (S), another nutrient element, is an essential building block of protein and Chl. Rhizobium also requires it during nitrogen fixation in legumes. Nikiforova et al. [62] demonstrated that the reduced biomass, protein, Chls, and total RNA resulted from a lower metabolic activity under limited S supply. Declined levels of S-containing metabolites such as cysteine and GSH were additional indicators of the limited input of S [84]. Katahira & Ashihara [85] observed the increased catabolism of purine and pyrimidine bases in S-deficient plants. Sulphur-starved *Arabidopsis thaliana* utilises the ureide pathway to restore nitrogen, sensed as excess under sulphur/nitrogen imbalance.

### 3.7. Ultraviolet-C

Solar radiation is a complex combination of ultraviolet (UV), visible and infrared rays. The UV radiation spectra emitted by the sun is sub-categorised into three major regions, namely UV-C, UV-B, and UV-A, having 100–280 nm, 280–315 nm, and 315–400 nm absorbance ranges, respectively [63]. Owing to the shortest wavelength, UV-C is the most energetic one. Photosynthetic apparatuses are identified as prime targets of UV-C imposed damages, thus remarkably affecting photosynthetic efficiency. Plants often show morphological responses when they have adjusted to stress situations. The curling of the leaf margin is one such strategy to reduce the rate of transpiration. Allantoin-treated plants witnessed a reduced transpiration rate with a simultaneous rise in the shoot fresh weight (Figure 2) and leaf water content. Dawood et al. [63] recorded phenotypic abnormalities and growth reduction in *Solanum lycopersicum* after UV-C (0.6 Wm^−2^) exposure. However, the treatment of 100 nM allantoin reversed these damages. Exposure to UV-C triggered photooxidative damages and elevated the levels of MDA and MG. 

On the other hand, exogenous allantoin exhibited enhanced levels of enzymatic and non-enzymatic antioxidants to eliminate these toxic products and maintain redox homeostasis inside cells. In addition, allantoin improved phenylalanine ammonia lyase activity, including anthocyanin, α-tocopherol, flavonoid, and total phenolic contents [63]. Moreover, enhancements in Chl and carotenoid content were observed after allantoin treatment, suggesting their important roles in strengthening plants’ photosynthetic efficiency under UV-C stress (Figure 2). Studies have revealed that the treatment of plants with allantoin prior to UV-C exposure is a more effective way to mitigate adverse impacts than the post-exposure treatment. Allantoin exhibits its photoprotective action by provoking the plant antioxidant system and accumulating UV-C-absorbing compounds [63].

Soltabayeva et al. [64] explored wild-type and an Atxdh1-knockout mutant defective in the XDH1 of *Arabidopsis thaliana* to examine the role of degraded purine metabolites (allantoin) in response to a combination of stressors, i.e., wounding and/or UV-C stress (0.15 J m^−2^ for 49 s) (Table 1). Ureides accumulated similarly against both UV-C irradiation and wounding in *Arabidopsis thaliana* leaves. The regulation of purine metabolism is controlled at the transcript level. Wounding and UV-C irradiation augmented the transcripts levels of ureide producers XDH and UOX, whilst transcript levels of ureide-degrading enzymes ALN and AAH were decreased. Mutant plants defective in XDH1 revealed accelerated senescence symptoms, including a yellowing of the leaves, reduced Chl content, and low levels of chloroplast proteins. A decline in Chl concentration in old leaves of Atxdh1 compared to the wild type has been associated with strong decreases in the levels of the large and small subunits of RuBisCo. The UV-C irradiation and double-wounding of middle leaves also decreased the Photosystem-II oxygen evolving complex and Glutamine Synthase 2 in the Atxdh1 mutant. Senescence was concomitant with higher expressions of the senescence marker genes *WRKY53* and *SGN1*. Mutants with impaired ureide accumulation were prone to increased ROS, MDA, electrolyte leakage, and tissue death (Table 1). Nevertheless, wild-type plants could withstand with the negative impacts of stressors, indicating the role of allantoin in providing antioxidants to decrease ROS levels and tissue mortality in the damaged leaves, and support membrane integrity. Ureides play a dual role under stress conditions. They function and remobilise from stress-damaged tissues to accumulate in damaged leaves to initially provide oxidative-stress scavenging, and then later support the growth of young leaves during the recovery period [79].

## 4. Hormonal Regulation of Allantoin Activation

Adaptive responses of plants to various biotic/abiotic factors are regulated by a fine network of signalling cascades of varied phytohormones. The cross talks between phytohormones are essential to modulate plant responses. Moreover, stress resistance may also depend on the generation and transmittance of hormone signalling that regulates the cellular metabolism [86]. Abscisic acid may show additive, antagonistic or synergistic relationships with other phytohormones, such as salicylic acid (SA) and ethylene. Moreover, ABA, in particular, has long been known to positively regulate jasmonic acid (JA) responses [13].

The accumulation of stress hormones ABA and JA in allantoin mutants and/or due to the application of exogenous allantoin anticipated a promising association between allantoin and the hormonal signalling cascade [10,87]. A correlation between allantoin and the ABA metabolism demonstrated that the allantoin stimulated the accumulation of ABA via the transcriptional up-regulation of two crucial enzymes: 9-cis-epoxycarotenoid dioxygenase (NCED3) and β-glucosidase 1 (BG1) [10]. Enzyme NCED3 oxidatively cleaves 9-cis-violaxanthin and/or 9-cis-neoxanthin to produce xanthoxin, which acts as a precursor for ABA biosynthesis, following multiple intermediate steps, whereas BG1 catalyses the conversion of the bound form of ABA into free form to stimulate its further responses [88] (Figure 3). Studies on ABA-deficient mutants aba2-1 (abscisic acid-deficient 2-1) and bglu18 (β-glucosidase 18) with affected de novo ABA biosynthesis and BG1 catalysed deconjugation of ABA glucose esters, respectively, suggested that the allantoin-mediated stimulation of ABA requires functional ABA biosynthesis and its deconjugation pathways [10]. This was supported by further experimentation, where a 2.5-fold enhancement in the levels of ABA was observed in *Arabidopsis thaliana* seedlings after exogenous allantoin application. In another study, an enhanced ABA concentration was concomitant with the up-regulation of ABA-responsive genes *NCED3*, *RD29A*, *RD29B*, *RD26,* and *COR15A* under osmotic and drought stress. Microarray studies revealed 2–16-fold higher transcript levels of stress-inducible genes in mutant plants [10]. However, no effect of allantoin on ABA production was evident in ABA-deficient aba2-1 or BG1-deficient bglu18 mutants. Allantoin is supposed to be directly involved in the mechanism of BG1 activation and the co-localization of BG1, and allantoin in ER plays a critical role in eliciting allantoin-mediated stress responses.

Allantoin has the potential to modulate the stress response at the gene expression level through the activation of metabolic pathways leading to ABA accumulation. The constitutive accumulation of allantoin in ALN mutants also triggers the JA pathway through ABA interaction. Studies have shown that the JA and its methyl esters [methyl jasmonate (MeJA)] initiate the signal transduction pathway and the expression of genes in response to external stimuli, consequently enhancing the resistance to abiotic stresses. Marker genes of JA metabolism and signalling, namely *13-LIPOXYGENASE* (LOX) and *JAZ PROTEINS*, acted in parallel to allantoin accumulation. The jasmonic acid signalling pathway functions via two oppositely acting branches, the MYC branch and the ERF branch, which are co-regulated by ABA and ethylene, respectively. The MYC branch, under the control of basic helix-loop-helix leucine zipper transcription factors MYC2/3/4, is active against herbivore feeding and mechanical wounding. The ERF branch, under the control of APETALA2/ERF domain transcription factors ERF1 and OCTADECANOID RESPONSIVE ARABIDOPSIS AP2/ERF 59, is active against necrotrophic pathogens. The ALN mutation has contradictory effects on these two major branches, triggering the MYC branch while repressing the ERF branch. Correspondingly, Takagi et al. [87] observed that the higher allantoin concentration stimulates downstream MYC marker genes, namely NAC transcription factors *(NAM/ATAF/CUC)* in *Arabidopsis thaliana*. Microarray studies of the ALN mutant revealed significant increases in JA metabolic enzymes such as LOX3, LOX4, allene oxide synthase, oxophytodienoate reductase, *JASMONATE ZIM-DOMAIN* (JAZ3), JAZ10, and MYC2. The activation of MYC2 activates JA signalling components including JAZ and JASMONATE-ASSOCIATED MYC2-LIKE1 repressor proteins.

The crosstalk between ABA and JA pathways is mediated through the transcriptional regulation of MYC2, which is considered to be a key regulator of the JA signalling pathway. It is a connecting link that integrates ABA and JA signalling (Figure 3). While activating the MYC branch, transcriptional regulator *MYC2* represses the downstream representatives of ERF branch markers, PLANT DEFENSIN 1.2 (PDF1.2a and PDF1.2b) and ORA59. In addition, MYC2 suppresses SA-dependent defences against biotrophic pathogens and participates in the biosynthesis of JA and anthocyanin, JA-induced root growth inhibition, and oxidative stress tolerance [87]. Abscisic acid acts upstream of the JA pathway to regulate *MYC2* activation, and allantoin activates JA responses by enhancing the expression of *MYC2*. Jasmonic acid and its metabolites are influenced by the ABA signalling pathway. 

Hormonal crosstalk between ABA and JA resulted in the enhanced production of JA, which strengthens resistance against biotic stresses like the infection of pathogens and herbivores [87]. The ALN mutant had a 2-fold increase in the level of JA and a 6-fold increase in the concentration of JA conjugated with amino acid isoleucine (JA-Ile) [87]. Jasmonic acid-responsive gene expressions in mutants JA-insensitive jar1-1 (JAR1 catalyses the formation of biologically active JA-Ile) and MYC2-deficient myc2-3 showed no response, even in the presence of allantoin. This shows that allantoin activity relies on the formation of bioactive JA-Ile. Both JA-insensitive and ABA-deficient mutants were insensitive toward allantoin levels (either externally supplied or internally accumulated) due to ALN mutation. Such findings reveal that allantoin activates JA responses through ABA-dependent MYC2 activation. Allantoin positively enhances the ABA level, activates stress-related genes, and confers abiotic stress tolerance on plants [10].

The role of allantoin is not limited to transport and storage as a nitrogen compound, and it also possesses ecological functions. A recent study revealed that allantoin in the root exudates is a key mediator of kin recognition in indica-inbred and indica-hybrid *Oryza sativa* cultivars [89]. Root-secreted allantoin acts as a growth-stimulating compound and activates the expression of genes involved in auxin biosynthesis, namely *OsYUCCAs* and its transport, via the up-regulation of *OsPIN1*, *OsPIN2*, and *OsAUX1* genes, thereby promoting futile root growth in non-kin species grown near *Oryza sativa* cultivars. An increase in root growth with decreasing relatedness was correlated with allantoin levels in the root exudates.

## 5. Conclusions and Future Prospects

Abiotic stress-induced toxicity is one of the major problems in the environment, and it adversely affects agricultural production and, consequently, food security. Plants have innate defensive machinery and regulatory networks to respond to toxicity. They have evolved to diversify and expend their metabolism to perform dual functions of growth and survival under stressful environments. The scientific observations on several plant species exposed to abiotic stresses demonstrated that the ureides, especially allantoin, are key players in determining their comparative resistance. In general, abiotic stress provokes the over-accumulation of ROS, and consequently oxidative stress situations. It has been observed that, along with the aggregated metal ions, high concentrations of ROS affect plants negatively. 

Furthermore, ROS react with lipids, proteins, and nucleic acids and initiate a sequence of damaging reactions, which culminate in cellular dysfunction. However, various enzymatic and non-enzymatic antioxidants, together with allantoin, have been identified to be involved in ROS detoxification and stress alleviation. Allantoin is a ubiquitous metabolite formed by the breakdown of purine bases. Various enzymes, such as oxidases, are common in all organisms that may contribute to allantoin biosynthesis and play a role as stress-protectant molecules in abiotic stress conditions.

To verify the mechanism of the stress-induced synthesis of allantoin, and it’s mediated tolerance in plants, various knockout, knockdown, and stress-inducible mutants/transgenic varieties (RD29A, ALN/aln-2) have been developed by targeting different genes and regulating the level of allantoin in plants. Additionally, several plant species exhibit tolerance behaviour, naturally. Basic elevation in the activities of antioxidants (SOD and APX) in aln-3 mutants prepares plants for ROS accumulation by increasing the ability to detoxify these compounds.

A number of plants, including *Arabidopsis thaliana*, have been shown to share a common tolerance strategy with concomitant allantoin accumulation to various abiotic stresses such as salt, drought, HM, and irradiation. Owing to the important role of allantoin in plant defence, it has become essential to understand whether the ureide pathway is the only route for allantoin biosynthesis or whether any other alternative or interconnected pathways exist in the cell system. Allantoin content has also been utilised for screening stress-tolerant varieties of *Glycine max*, *Cicer arietinum*, and *Oryza sativa*, and can be used as a stress marker in these species. For a deeper understanding of the regulatory roles of allantoin, we also need to understand the transcriptional and translational control of genes involved in allantoin biosynthesis and transport and its regulation under different stress conditions. It is also necessary to investigate regulation aspects to understand which molecular switches control flux through the pathway. A possible interaction between BG1 and allantoin and its regulation could be elucidated through in vitro experiments targeting BG1 recombinant proteins. 

Additionally, microarray-based studies will be required to understand the expressions of different genes involved in boosting the plant defence system in relation to allantoin. Synergistic or antagonistic interaction of allantoin with other antioxidative systems is yet to be revealed. Molecular studies may be conducted to understand the complete interplay between allantoin and the activation of stress tolerance pathway(s) in plants. A detailed analysis and further experimentation regarding the ureide build-up on physiological, biochemical, and molecular levels and the in-depth characterization of mutants may unveil the possible connection between allantoin and stress responses. Cellular accumulations of allantoin play several additional functions which still need to be unveiled.

## Figures and Tables

**Figure 1 plants-12-03059-f001:**
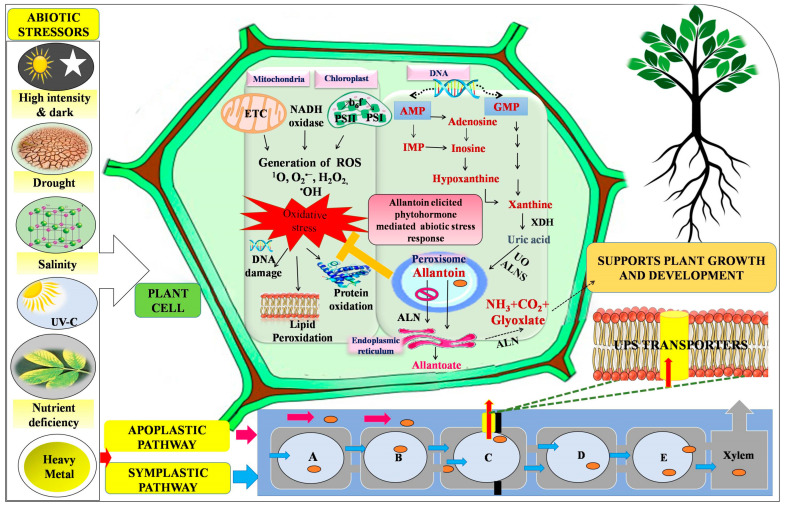
Diagrammatic representation of biosynthesis and transportation of allantoin under oxidative stress conditions. Disturbance in the normal cellular metabolism of plants due to abiotic stress conditions provokes the over-accumulation of reactive oxygen species (ROS), leading to oxidative stress, which is detrimental to vital cellular components. In order to overcome stress responses, plants have developed defensive mechanisms through alterations in the purine catabolism pathway. (A: epidermis; B: cortex; C: endodermis; D: pericycle; E: vascular bundle; AMP: adenosine monophosphate; ALN: allantoinase; ALNS: allantoin synthase; ER: endoplasmic reticulum; GMP: guanosine monophospahte; IMP: inosine monophospahte; UO: uricase; XDH: xanthine dehydrogenase).

**Figure 2 plants-12-03059-f002:**
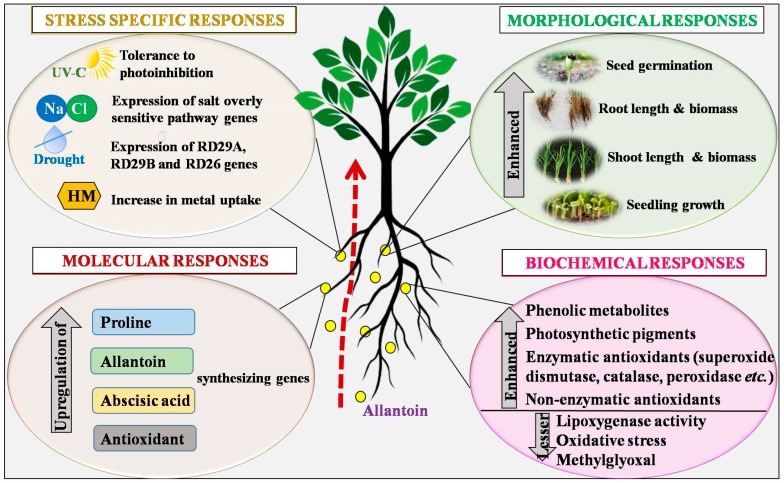
The mechanistic role of allantoin in alleviating the adverse impacts of abiotic stressors. Exogenously applied or innately synthesised allantoin exhibits several physio-biochemical, molecular and stress-specific responses under stress conditions. [RD29A, RD29B, and RD26 (response to desiccation)].

**Figure 3 plants-12-03059-f003:**
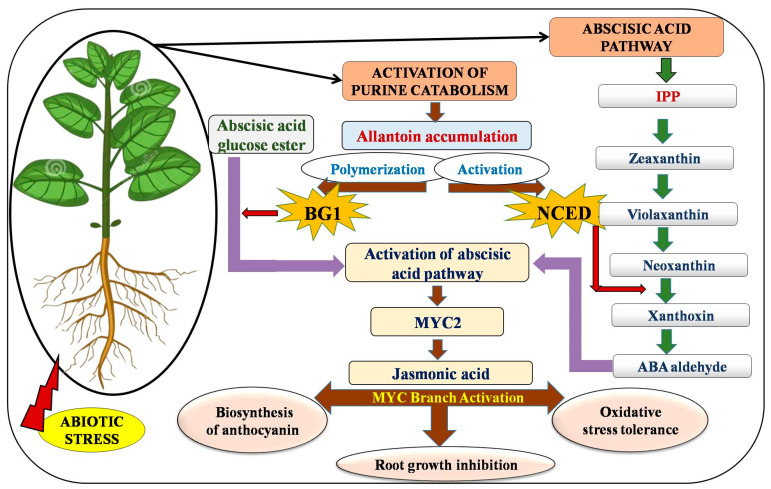
Allantoin-elicited phytohormone-mediated abiotic stress response in plants. Allantoin increases the level of abscisic acid (ABA) in plant cells either by the activation of 9-cis-epoxycarotenoid dioxygenase (NCED) or throughβ-glycosidase homolog (BG1) hydrolytic conversion of inactive bound ABA into free form of ABA. Free ABA activates the jasmonic acid (JA) signalling pathway via the regulation of MYC2 transcription factor. Jasmonic acid functions to promote the biosynthesis of anthocyanin, root growth inhibition, and tolerance to oxidative stress.

## Data Availability

No new data were created or analyzed in this study.

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
