# Peer review of "Allantoin: A Potential Compound for the Mitigation of Adverse Effects of Abiotic Stresses in Plants"

_plants, 2023, doi:10.3390/plants12173059_

Round 1

Reviewer 1 Report

Purine catabolism acts as an inherent defensive mechanism against abiotic stressors and plays pivotal role in stress acclimatization of plants. Intermediate metabolite of purine catabolism, allantoin, compensates for soil nitrogen deficiency due to low carbon/ nitrogen ratio, thereby maintaining nitrogen homeostasis and supporting plant growth and development. In this study, the metabolic regulation of allantoin as photoprotectant, osmoprotector and nitrogen recovery agent and its possible role in reducing the effects of abiotic stress on plants were systematically discussed by summarizing previous studies. This article is well organized, with reasonable arguments and novel references, which in line with the requirements of the magazine. However, lots of writing errors should be revised. Following are some examples. You may better revise the manuscript by native speaker.

L11. “abiotic tresses”?

L14. Change “provide” to provides”.

L18. “abiotic stressors”?

L20. “carbon/ nitrogen” should not contain blank after /.

L22. “revealed”?

L29. “knockdown/ knockout” should not contain blank after /. Thoroughly check the manuscript, in most conditions, it is similar to and/or.

L44. “however;”?

L88. In figure 1, please uniform the writing patterns, take “ABIOTIC STRESSORES”, “Plant Cell”, “Apoplastic pathway” as examples, why they are showed as different writing patterns. Also check other figures.

Lots of writing errors should be revised. You may better revise the manuscript by native speaker.

Author Response

Gary Cao, Editor (Plants-MDPI)

Dear sir:

Reviewer 2 Report

The review (Allantoin: A potential compound for mitigation of adverse effects of abiotic stresses in plants ) is interesting and well written. The figures, tables and content were well organized, and minor revisions need to be done before the paper can be accepted. My comments are in the attachment.

Minor editing of English language required

Author Response

(The authors gave the same response as above.)

Reviewer 3 Report

The authors reviewed the role of ALLANTOIN in the Abiotic stress tolerance in plants. The review was well organised and written well. However, the authors needs to put more effort in the illustrations.

1. The Fig. 1 is needs to be worked more to present interactive and to give a clear information. It is not well connected. Also, the description looks like a big explanation which is duplication. Give a clear short information how the figure helps to understand the hypothesis documented inside the text.

2. Fig 2 requires formatting. Uniform alighnment including font size, line space, etc will be make the figure a better appearance. Also P5CS1 looks odd in the list of molecular chaparons. Instead proline will be better.

3. Expansions of abbreviations need to mentioned at their first appearance. Few are missed. Example “Proton NMR-based metabolic profiling” in line 266.

Quality of language is quit good. However, minor improvement is required.

Author Response

(The authors gave the same response as above.)

Reviewer 4 Report

I am satisfied that common link between all organisms is confirmed in the paper basing on the urine metabolism , especially for allantoin as defensive agent. Authors gave good schemes  of the mechanisms. Allantoin is formed in mammalian embryo that explain its importance for earlier protection via the decompose of purine bases. Similar way seems to be common for all organisms, and authors may point this thing in the conclusion ( the end phrase or something else).  Very interesting facts are here that various enzymes common for all organisms may participate in the formation of allantoin, for example oxidases. This idea may be also marked. Role of phytohormones is not so significant.

Author Response

(The authors gave the same response as above.)

Round 2

Reviewer 1 Report

Nice work for the revision